# Void Suppression in Glass Frit Bonding Via Three-Step Annealing Process

**DOI:** 10.3390/mi13122104

**Published:** 2022-11-29

**Authors:** Yifang Liu, Junyu Chen, Jiaxin Jiang, Gaofeng Zheng

**Affiliations:** 1Department of Instrumental and Electrical Engineering, Xiamen University, Xiamen 361102, China; 2Shenzhen Research Institute, Xiamen University, Shenzhen 518000, China; 3School of Mechanical and Automotive Engineering, Xiamen University of Technology, Xiamen 361024, China

**Keywords:** glass frit, void suppression, annealing, parameter optimization

## Abstract

In this work, void formation was systematically observed for the glass frit bonding technique as a function of the annealing temperature, annealing time, and annealing ambient. High annealing temperature and long annealing time were adopted to reach the maximum heat flux to avoid voids/bubbles. As demonstrated in the experiments, the voids appearing during glass frit bonding are related to the quantity of byproducts from the combustion of organic matter. The experimental results indicate that solely in air, under vacuum, or annealed for short time, the combustion products cannot be fully degassed, and voids occur. It was shown that the alternating three-step conditioning process including glass liquid forming in air, bubble removal under vacuum, and void filling-up in air can lead to void-free and uniform wafer bonding. The glass frit bonding samples with lots of voids/bubbles were compared to the ones without any defects.

## 1. Introduction

Glass frit bonding is a viable option for a wide range of microelectromechanical (MEMS) devices, such as nanofluidic channels type leak assembly [1], pressure sensors [2,3], micro-mirrors [4] and RF MEMS devices [5] etc. It finds applications in low cost and time effective hermetic package fabrication process [6]. It is also capable to cover a few micrometer surface steps, which works for simple planar feed through structures [7]. Compared with direct wafer bonding, tiny particles do not cause voids because the glass frit flows around the particles in glass frit bonding. However, voids initially exist in the glass paste in the untreated glass cord resulting from the evaporation of solvents and binders. The voids not only result in leakage pathways, but also reduce the effective contact area of the bonding interface. Therefore, the voids deteriorate the sealing performance and decrease the bonding strength. A void-free interface is the key to the improvement of bonding strength, which has significant impact on the reliability of the fabricated devices as wafer bonding processing is often a middle step for device manufacturing [8].

The pre-sintering process is crucial for obtaining a void-free sealing layer [9]. The organic materials in the glass frit cannot volatilize completely if the peak annealing temperature is too low. Therefore, the residual organic substance will continue to release some gases leading to lots of pores after pre-sintering and bonding [10]. When the bonding temperature is 400 °C, the glass paste is not completely melted, which leads to an uneven bonding interface. After bonding, there are large holes in the glass paste layer. However, when the bonding temperature is increased to 500 °C, the bubble size inside the glass paste decreases significantly [10]. The excessive liquid of glass frit resulting from over burning by too high a bonding temperature causes irregular solidification in the cooling process, which leads to the formation of voids. Insufficient bonding pressure also causes the occurrence of voids at the uneven bonding interface [11]. For the native silicon substrate and a cap wafer without an intermediate layer such as oxide and nitride, the formation of small spherical lead precipitations in the fracture plane results in a large number of voids [12,13]. Ser Choong Chong et al. [14] found that after annealing the oxide layer on the cap layer before bonding, there were almost no holes in the interface between the glass paste ring and the cap layer. Therefore, it is considered that the release of air from the oxide layer in the bonding interface leads to the formation of pores. A literature search revealed that other researchers believe void formation is an issue that must be addressed when using glass frit to create a hermetic seal, and that fine control of the bonding conditions is needed to achieve a void-less bond [15,16].

In this work, a stable optimized alternating three-step annealing process, which included the forming of glass liquid in air, the elimination of bubbles under vacuum, and the filling-up of pores in air was introduced to suppress the formation of voids. Pull tests were carried out to examine the bonding strength. The leaking rate of the bonding pairs was obtained by Helium leak tests after the proposed annealing process and bonding.

## 2. Experimental Approach

In our experiments, transparent Pyrex 7740 glass was used as the cap wafer to inspect the state of the voids/bubbles. The commercially available silicon wafer (4 inch) substrate had one side polished. The glass paste purchased from KOARTAN (Randolph, NJ, USA) was 5643w. The CTE of the glass frit was 6 ppm/℃. Before thermal conditioning and bonding processing, the glass paste was screen printed on the silicon substrate by the semiautomatic Marabu MS-300F screen process press. The optimized pre-sintering was carried out in a tube furnace (TF55035C, LINDBERG/BLUE, Waltham, MA, USA) after screen printing. Annealing time, annealing temperature, and annealing ambient were set step by step to attain the best glass frit layer without any voids. Finally, both the cap wafer and the substrate wafer were brought into the vacuum bonder (AWB04) for 60 min. The bonding temperature and pressure were 450 °C and 10 atm, respectively. The cooling rate was 2.5 °C/min.

The morphologies of the frit paste were investigated by a microscope system (Mitutoyo, Kanagawa, Japan). The cross-sectional views were obtained by a dicing machine (ZSH406, Shenyang Academy of Instrumentation, Shenyang, China) followed by manual polishing. The state of the voids/bubbles was inspected with scanning electron microscopy (SEM 1530, LEO, Germany; S-4800, Hitachi, Tokyo, Japan) and a CCD camera (UI-2250SE-HQ, IDS, Obersulm, Germany). The bonding strength was measured by a pull test using an electronic testing machine (SHIMADZU, Kyoto, Japan). The leak rate of the glass frit bonding structure was performed on a Helium leak detector (ASM 192T, Pfeiffer, Aßlar, Germany). All tests were performed in an air laboratory. For each different pre-sintering condition, we manufactured three bonding pairs because of the high bonding cost. We performed the observation of the morphology of the glass frit on all the three bonding pairs for each annealing condition. The effects of the annealing temperature, time, and atmosphere on the removal of the bubbles from the morphology of the glass frit in each annealing process were compared.

## 3. Results and Discussions

### 3.1. Morphology of Glass Frit after Pre-Sintering

The applied experimental conditions are listed in Table 1. To distinguish the formation of the voids/bubbles in air from the ones under vacuum ambient, Case 1 and Case 2 were carried out. Because the fining temperature must be higher than the melting temperature during the melting and fining of glass, the annealing temperature was increased to 450 °C in Case 3. In Case 4 and Case 5, annealing was done in air and vacuum alternately. In Case 5, the annealing time was further optimized. In consideration that the void formation mechanism is a function of total heat flux [17], Case 6 was proposed to increase the total amount of heat flux to the sample by increasing the annealing temperature in the third step to eliminate voids.

Figure 1 shows the size, count, and distribution of the voids at the surface of the layer of the glass frit. In Figure 1a, we can see that many cracks appear, but there are no voids/bubbles on the surface. Figure 1b shows that after being pre-sintered purely under vacuum for 60 min, a huge number of deep voids form at the surface. In Case 3, the glass frit was annealed in air ambient for 30 min to make the glass frit reflow and under vacuum for 30 min to exhaust the large quantity of releasing gases [18]. As can be seen in Figure 1c, a relatively clean and smooth glass frit layer was formed, and the voids were significantly smaller than those in Figure 1b. Figure 1d shows that annealing in air after bubble evacuation under vacuum leads to a significant decrease in the number of voids. An extended hold time from 30 min to 60 min further reduced the voids (Figure 1e). As can be seen in Figure 1f, a void-less and smooth frit layer was obtained [19].

The difference between Figure 1a,b is attributed to the slower diffusion of gas molecules to the surface in air ambient than under vacuum. In pure annealing vacuum, bubbles hidden inside the glass liquid are squeezed up to the surface. When the bubbles burst, holes are formed. If the holes are not filled up in time, there will be a large number of voids after cooling. These voids function as the starting points for peelings or breakages to cause defective cracks. Comparing Figure 1a with Figure 1b, the important impact of the annealing ambient on the void’s formation is obvious. Comparing Figure 1c with Figure 1b, we can see that thermal annealing under vacuum followed by annealing in air potentially decreases the voids as gases were squeezed out more effectively. However, lots of little voids and cracks are still distributed on the surface of the glass frit. When the annealing time and temperature increase simultaneously, the viscosity of frit decreases and the molten glass frit flows faster, so the holes were filled up. The whole thermal bonding process with no visible voids is shown in Figure 2, compared with initial thermal conditional profile provided by the manufacturer of the glass frit. The bonding parameters are shown in Table 2.

### 3.2. Morphology of Glass Frit after Pull Test

The voids in the fractured interface after the pull test are demonstrated in Figure 3. Figure 3 was used to inspect the state of the voids at the interface of the glass—glass frit—silicon bonding structure. There were many voids inside the glass frit layer after bonding with annealing only in air, as shown in Figure 3a–c. The size and number of bubbles in Case 1, Case 2, and Case 3 were much alike. Figure 3d suggests that some of the voids were filled with the melting glass paste, the number and length scale of the voids decreased. Figure 3e,f demonstrate that high annealing temperature and long annealing time greatly reduced the voids. The blue lines in the Figure are the cracks left along the relatively fragile positions of the bubbles during tensile test.

### 3.3. Morphology of Glass Frit after Bonding

Comparing Figure 1a, Figure 3a and Figure 4a, no voids were found at the surface of the layer of glass frit (Figure 1a), but a large number of voids were buried inside the glass frit as seen in Figure 3a and Figure 4a. Figure 1b, Figure 3b and Figure 4b show a large number of voids or bubbles at the surface of or inside the glass frit. From Figure 1c–f, Figure 3c–f and Figure 4c–f, the reduction of the number of voids can be explained by the merging of growing voids and the diffusion of gas byproducts in the alternating vacuum and air atmosphere conditions.

### 3.4. Bonding Strength and Leak Rate

Voids are undesirable in any packaging process as they might reduce the bonding strength, which affects the long-term reliability of the devices. For comparison, bonding pairs were made with initial thermal conditioning purely in air and optimized alternating three-step annealing process followed by the same bonding process. Pull tests [20] for the diced chips without any voids and those with some voids were carried out to determine whether the optimized alternating three-step annealing process would improve the bonding strength and decrease the leak rate.

The effect on bonding strength of the different thermal conditioning processes is shown in Figure 5. The bonding strength of the bonding pairs with the alternating three-step annealing process is the highest in all the Cases. Compared with the initial case, the mean bonding strength was increased from 10.2 MPa to 19.1 MPa by the optimal annealing process. Helium leak tests [21] were done on both three randomly selected bonding pairs thermally conditioned purely in air and three randomly selected assemblies made with the optimized alternating three-step annealing process. Table 3 shows that the measured leak rate of the three samples with annealing purely in air was 5.9 × 10^−8^, 6.2 × 10^−7^, and 6.2 × 10^−7^ atm cc/s, respectively, which is above the specification limit of MIL-STD883E of 5 × 10^−8^ atm cc/s. Under the optimized alternating three-step annealing process, the Helium leak value was 2.8 × 10^−8^, 3.3 × 10^−8^, and 3.5 × 10^−8^ atm cc/s, respectively, all of which are smaller than the standard value.

## 4. Conclusions

For the optimization of the pre-sintering process to suppress the formation of voids/bubbles in the glass frit bonding technology, the influence of annealing temperature, annealing time, and annealing ambient, were investigated. The optimized three-step pre-sintering process included forming glass liquid in air at 450 °C for 30 min, bubble evacuation under vacuum at 450 °C for 30 min, and void filling-up in air at 500 °C for 60 min. This process can suppress the formation of voids and increases the bonding strength from 10.2 MPa to 19.0 MPa. Leak test results showed that samples without any voids provide better hermetic performance (lower than 5 × 10^−8^ atm cc/s).

## Figures and Tables

**Figure 1 micromachines-13-02104-f001:**
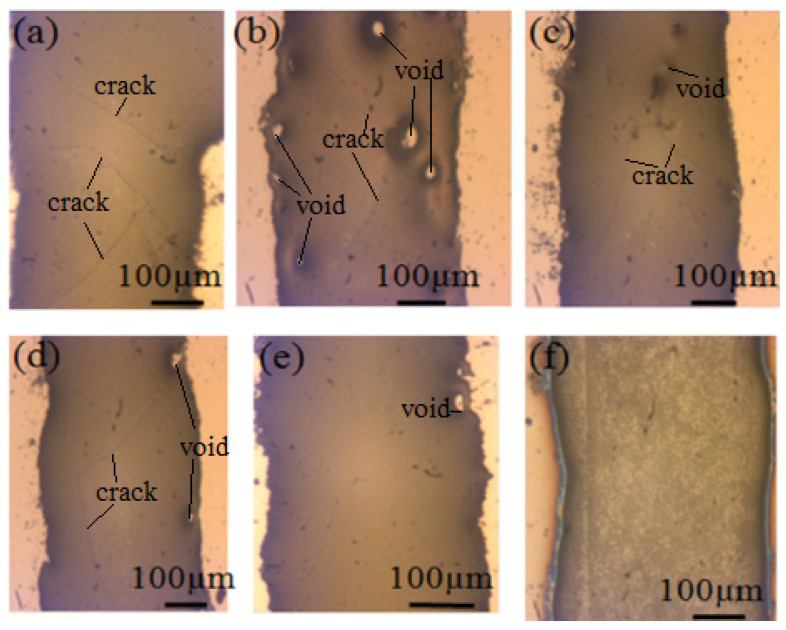
CCD images of the glass frit surface of different Cases after pre-sintering: (**a**,**b**) Annealing in air and under vacuum at 400 °C for 60 min, respectively; (**c**) Annealing in air at 450 °C for 30 min and annealing under vacuum at 450 °C for 30 min; (**d**,**e**) Annealing in air at 450 °C for 30 min and annealing under vacuum at 450 °C for 30 min and annealing in air at 450 °C for 30 min and for 60 min, respectively; (**f**) Annealing in air at 450 °C for 30 min and annealing under vacuum at 450 °C for 30 min and annealing in air at 500 °C for 60 min.

**Figure 2 micromachines-13-02104-f002:**
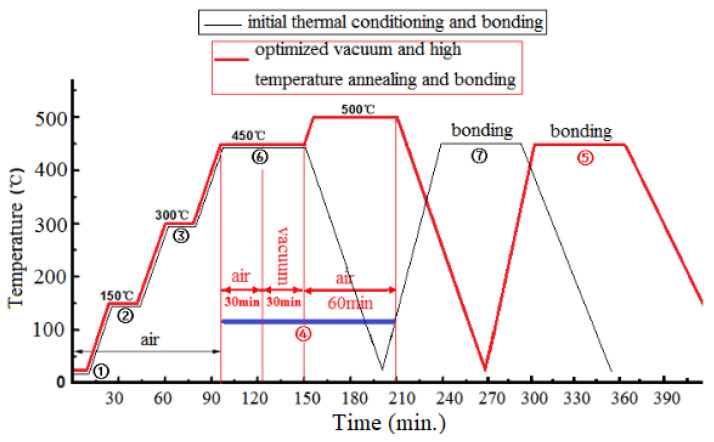
The thermal profile of glass frit bonding: ① After screen printing; ② Removal of the organic solvent; ③ Removal of the organic binder; ④ Optimized alternating three-step annealing process; ⑤, ⑦ Bonding; ⑥ initial pre-sintering (purely in air).

**Figure 3 micromachines-13-02104-f003:**
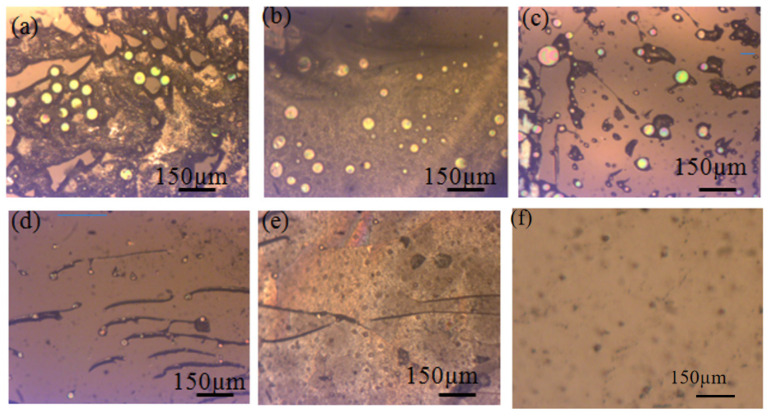
The voids in the fractured interface after the pull test: (**a**,**b**) Annealing in air and under vacuum at 400 °C for 60 min, respectively; (**c**) Annealing in air at 450 °C for 30 min and annealing under vacuum at 450 °C for 30 min; (**d**,**e**) Annealing in air at 450 °C for 30 min and annealing under vacuum at 450 °C for 30 min and annealing in air at 450 °C for 30 min and for 60 min, respectively; (**f**) Annealing in air at 450 °C for 30 min and annealing under vacuum at 450 °C for 30 min and annealing in air at 500 °C for 60 min.

**Figure 4 micromachines-13-02104-f004:**
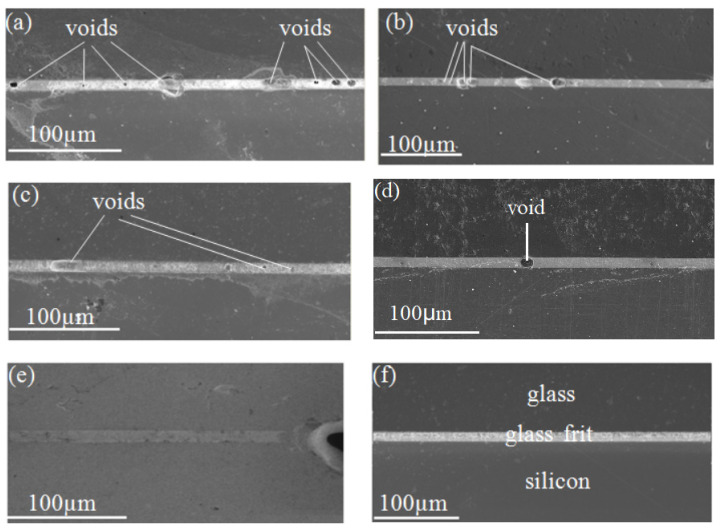
Cross sectional SEM image of glass frit bonds: (**a**,**b**) Annealing in air and under vacuum at 400 °C for 60 min, respectively; (**c**) Annealing in air at 450 °C for 30 min and annealing under vacuum at 450 °C for 30 min; (**d**,**e**) Annealing in air at 450 °C for 30 min and annealing under vacuum at 450 °C for 30 min and annealing in air at 450 °C for 30 min and for 60 min, respectively; (**f**) Annealing in air at 450 °C for 30 min and annealing under vacuum at 450 °C for 30 min and annealing in air at 500 °C for 60 min.

**Figure 5 micromachines-13-02104-f005:**
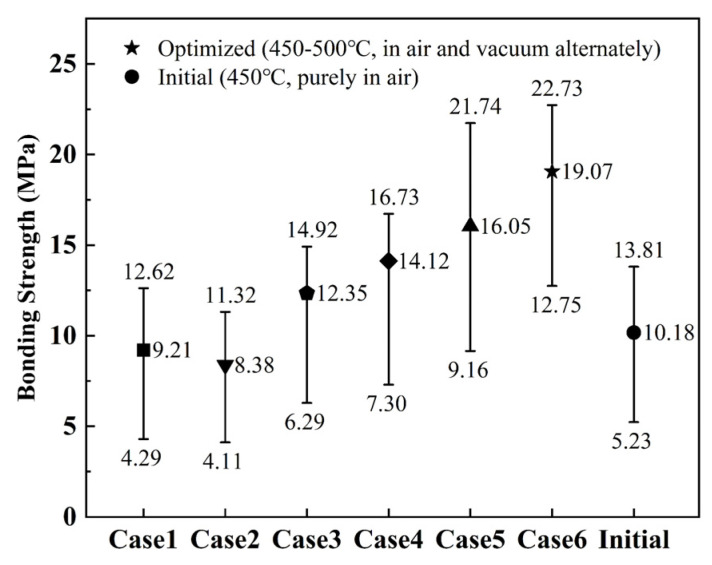
The effect on bonding strength of different thermal conditioning process.(**Case1**, **Case2**) Annealing in air and under vacuum at 400 °C for 60 min, respectively; (**Case3**) Annealing in air at 450 °C for 30 min and annealing under vacuum at 450 °C for 30 min; (**Case4**, **Case5**) Annealing in air at 450 °C for 30 min and annealing under vacuum at 450 °C for 30 min and annealing in air at 450 °C for 30 min and for 60 min, respectively; (**Case6**) Annealing in air at 450 °C for 30 min and annealing under vacuum at 450 °C for 30 min and annealing in air at 500 °C for 60 min; (initial) Annealing in air at 450 °C for 30 min.

**Table 1 micromachines-13-02104-t001:** Pre-sintering Cases.

Case	Annealing Temperature (°C)	Annealing Time (min)	Annealing Ambient	Result
1	400	60	Air	Figure 1a
2	400	60	Vacuum	Figure 1b
3	450	60	Air (30 min) and Vacuum (30 min)	Figure 1c
4	450	90	Air (30 min) and Vacuum (30 min) and Air (30 min)	Figure 1d
5	450	120	Air (30 min) and Vacuum (30 min) and Air (60 min	Figure 1e
6	450–500	120	Air (30 min) (450 °C) and Vacuum (30 min) (450 °C) and Air (60 min) (500 °C)	Figure 1f

**Table 2 micromachines-13-02104-t002:** Bonding Parameters.

Bond Temp (°C)	Hold Time (min)	Bond Pressure (atm)	Cooling Rate (°C/min)	Bond Ambient
450	60	10	2.5	Vacuum

**Table 3 micromachines-13-02104-t003:** The leak rate of initial and optimized bonded pairs.

Bonded Pairs’ No.	Leak Rate of Initial Bonded Pairs (atm cc/s)	Leak Rate of Optimized Bonded Pairs (atm cc/s)
1	5.9 × 10^−8^	2.8 × 10^−8^
2	6.2 × 10^−7^	3.3 × 10^−8^
3	6.2 × 10^−7^	3.5 × 10^−8^

## Data Availability

Not applicable.

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
