# Peer review of "Void Suppression in Glass Frit Bonding Via Three-Step Annealing Process"

_micromachines, 2022, doi:10.3390/mi13122104_

Round 1

Reviewer 1 Report

The generation of bubbles in the glass frit bonding process is a very common problem. In this paper, appropriate process parameters have been found through experimental optimization. But the whole paper is like an experimental report, rather than a scientific manuscript. There are some questions that need to be explained:

1. line79-88.  Is there any theoretical basis for selecting these temperature and time parameters? 

2. Bonding bubble problem is a very common problem. Have other papers discussed this issue? Relevant references can be provided for comparison.

3. Table 1. Case 6. “+” is redundant.

4. Figure1. It is necessary to mark where the crack is and where the void is.

5. line116. Why the gases are squeezed out more effectively

6. line 122. Which manufacturer provided the parameters? According to this figure, whether it is understandable that your optimization is just add an annealing process in 500 ℃ air?

7. Figure 3.  Why the ruler is different in Figure 3d?

8. line 130. Figure 4 appears in Section 3.3. It is obviously inappropriate to discuss here. It is better to reorganize the paper structure again. 

9. Figure 5. The picture is not very clear.

10. English writing needs improvement, many statements are confusing and verbose. And many figures are explained repeatedly.

Author Response

Dear Editors and Reviewer:

Thank you for your letter and for the reviewers’ comments concerning our manuscript entitled “Void Suppression in Glass Frit Bonding Via Three-Step Annealing Process” (Manuscript ID: micromachines-2025940). Those comments are all valuable and very helpful for revising and improving our paper, as well as the important guiding significance to our researches. We have studied comments carefully and have made correction which we hope meet with approval. Revised portion are marked in red in the paper. The main corrections in the paper and the responds to the reviewer’s comments are as flowing:

The generation of bubbles in the glass frit bonding process is a very common problem. In this paper, appropriate process parameters have been found through experimental optimization. But the whole paper is like an experimental report, rather than a scientific manuscript. There are some questions that need to be explained:

  1. line79-88.  Is there any theoretical basis for selecting these temperature and time parameters? 

A logical starting point is o determine how the pre-sintering parameters will affect the bond quality when using the established bonding sequence. Pre-sintering is used to remove unwanted components in the glass frit prior to bonding. And a well controlled reflow would produce a better bond. Use the conditions indicated in Figure2 from KOARTAN as a basis and speculating that void formation was due to an incomplete burn-off organics.

  1. Bonding bubble problem is a very common problem. Have other papers discussed this issue? Relevant references can be provided for comparison.

When the bonding temperature is 400℃, the glass paste is not completely melted, which leads to the uneven bonding interface. After bonding, there are large holes in the glass paste layer. But when the bonding temperature is increased to 500℃, the bubble size inside the glass paste decreases significantly[10]. Ser Choong Chong et al. found that after annealing the oxide layer on the cap layer before bonding, there were almost no holes in the interface between the glass paste ring and the cap layer. Therefore, it is considered that the release of air from the oxide layer in the bonding interface leads to the formation of pores[14]. A literature search revealed that other researchers believe void formation is an issue that must be addressed when using glass frit to create a hermetic seal, and that fine control of the bonding conditions is needed to achieve a voidless bond. In order to minimize the bubbles in the case of vacuum in-line sealing, we tried to form a new type of frit glass.The main reason for the bubbles seems to be vaporization of PbO molecules because the vapor pressure of the moleculars increases exponentially with the vacuum level [15,16].

  1. Table 1. Case 6. “+” is redundant.

We are very sorry for our incorrect writing and we have made correction in the paper.

  1. It is necessary to mark where the crack is and where the void is.

Thank you very much for your guidance! We have marked where the crack is and where the void is in Figure1.

  1. Why the gases are squeezed out more effectively.

The molten glass paste layer constantly produces bubbles inside, making the internal pressure greater than the external vacuum environment pressure. The pressure difference pushes the bubbles to the surface more efficiently.

  1. line 122. Which manufacturer provided the parameters? According to this figure, whether it is understandable that your optimization is just add an annealing process in 500 ℃ air?

The 5643W sealing glass frit from KOARTAN was applied in our experiments. An ideal pre-sinter should produce a clean burn off all organics without causing the frit to reflow. Crucial pre-sintering parameters are temperature followed by hold time. Void formation is a function of total heat flux delivered. It might be possible to eliminatethe formation of voids through extremely fine control of pre-sintering process.We optimized the annealing process which includes annealing in air at 450°C for 30 minutes , annealing in vacuum at 450°C for 30 minutes, and annealing in air at 500°C for 60 minutes. But the initial annealing process was carried out purely in air at 450. for 60 minutes.

  1. Figure 3.  Why the ruler is different in Figure 3d?

We have made correction according to the Reviewer’s comments.

  1. line 130. Figure 4 appears in Section 3.3. It is obviously inappropriate to discuss here. It is better to reorganize the paper structure again. 

Considering the Reviewer’s suggestion, we have reorganized the paper structure.

  1. Figure 5. The picture is not very clear.

We have redrawn Figure5 clearly in the manuscript.

  1. English writing needs improvement, many statements are confusing and verbose. And many figures are explained repeatedly.

We have improved the writing and deleted the verbose statement.

Reviewer 2 Report

The authors demonstrated a method to suppress the void formation for the glass frit bonding. The results is convincing enough for the publication. However, I believe that it can be further improved:

(i) Have the authors tried other conditions, such as higher temperature and longer time annealing? The conditions that author presented in the manuscript is not really optimized and can be improved further, especially in the bonding strength. 

(ii) What is the bonding strength of other pre-sintering methods? It would be nice to include all the 6 cases in the same graph.

(iii) Some minor English grammar mistakes can be improved.

Author Response

Dear Editors and Reviewer:

Thank you for your letter and for the reviewers’ comments concerning our manuscript entitled “Void Suppression in Glass Frit Bonding Via Three-Step Annealing Process” (Manuscript ID: micromachines-2025940). Those comments are all valuable and very helpful for revising and improving our paper, as well as the important guiding significance to our researches. We have studied comments carefully and have made correction which we hope meet with approval. Revised portion are marked in red in the paper. The main corrections in the paper and the responds to the reviewer’s comments are as flowing:

The authors demonstrated a method to suppress the void formation for the glass frit bonding. The results is convincing enough for the publication. However, I believe that it can be further improved:

  • Have the authors tried other conditions, such as higher temperature and longer time annealing? The conditions that author presented in the manuscript is not really optimized and can be improved further, especially in the bonding strength. 

Thank you very much for your valuable comments! It could be seen from the above results that after optimizing the pre sintering process, a dense glass frit layer without holes was obtained. And the bonding results that meet the requirements of the project were also obtained. And due to the limitation of project funds, to further improve the bonding strength , we increased the bonding pressure to 15 atm. The bonding strength was increased to 26.76 MPa.

  • What is the bonding strength of other pre-sintering methods? It would be nice to include all the 6 cases in the same graph.

Thank you for your guidance! We have made a new Figure 5 which included all the 6 cases and the initial case (annealing purely in air at 450℃ for 60 minutes)in the same graph.

Figure 5. The effect on bonding strength of different thermal conditioning process.

  • Some minor English grammar mistakes can be improved.

We have improved the writing in the text.

Round 2

Reviewer 1 Report

thanks for the improved version of your paper on "Void Suppression in Glass Frit Bonding Via Three-Step An- 2 nealing Process". The authors  have addressed all points of my review and the paper is in my opinion now ready to publish